# Enhancing Si_3_N_4_ Selectivity over SiO_2_ in Low-RF Power NF_3_–O_2_ Reactive Ion Etching: The Effect of NO Surface Reaction

**DOI:** 10.3390/s24103089

**Published:** 2024-05-13

**Authors:** Nguyen Hoang Tung, Heesoo Lee, Duy Khoe Dinh, Dae-Woong Kim, Jin Young Lee, Geon Woong Eom, Hyeong-U Kim, Woo Seok Kang

**Affiliations:** 1Mechanical Engineering, KIMM Campus, University of Science and Technology (UST), Daejeon 34113, Republic of Korea; nhtung79@kimm.re.kr (N.H.T.); dwkim@kimm.re.kr (D.-W.K.); 2Semiconductor Manufacturing Research Center, Korea Institute of Machinery and Materials (KIMM), Daejeon 34103, Republic of Korea; hslee89@kimm.re.kr (H.L.); geonous0319@kimm.re.kr (G.W.E.); guddn418@kimm.re.kr (H.-U.K.); 3R&D Center, Naieel Technology, Daejeon 34104, Republic of Korea; khoedd@gmail.com; 4Department of Nano-Devices & Display, Korea Institute of Machinery and Materials (KIMM), Daejeon 34103, Republic of Korea; ljycj12@kimm.re.kr; 5Department of Physics, Chungnam National University (CNU), Daejeon 34134, Republic of Korea; 6Nano-Mechatronics, KIMM Campus, University of Science and Technology (UST), Daejeon 34113, Republic of Korea

**Keywords:** reactive ion etching (RIE), selectivity, Si_3_N_4_, SiO_2_, reaction surface

## Abstract

Highly selective etching of silicon nitride (Si_3_N_4_) and silicon dioxide (SiO_2_) has received considerable attention from the semiconductor community owing to its precise patterning and cost efficiency. We investigated the etching selectivity of Si_3_N_4_ and SiO_2_ in an NF_3_/O_2_ radio-frequency glow discharge. The etch rate linearly depended on the source and bias powers, whereas the etch selectivity was affected by the power and ratio of the gas mixture. We found that the selectivity can be controlled by lowering the power with a suitable gas ratio, which affects the surface reaction during the etching process. X-ray photoelectron spectroscopy of the Si_3_N_4_ and QMS measurements support the effect of surface reaction on the selectivity change by surface oxidation and nitrogen reduction with the increasing flow of O_2_. We suggest that the creation of SiO_x_N_y_ bonds on the surface by NO oxidation is the key mechanism to change the etch selectivity of Si_3_N_4_ over SiO_2_.

## 1. Introduction

Silicon nitride (Si_3_N_4_), a widely used silicon-based material in the contemporary semiconductor industry, has been implemented in complementary metal (oxide) devices as a gate spacer and diffusion barrier against alkali and copper ions [1]. It has recently been used as a sacrificial charge-trapping layer in three-dimensional (3D) vertical NOT AND (NAND) flash memory devices [2,3]. In these devices, Si_3_N_4_ is combined with silicon dioxide (SiO_2_), which is another dielectric widely used as a premetal, intermetal, tunneling oxide, and blocking oxide [4]. To integrate them into complex semiconductor devices for various applications, Si_3_N_4_ and SiO_2_ must be selectively removed without affecting the other constituents. For instance, the gate electrodes in 3D vertical NAND devices must be formed by selectively removing Si_3_N_4_ while minimally etching SiO_2_. However, the etching should be performed with excellent selectivity in self-aligned contact methods [5,6,7,8,9]. 

Particularly, 3D vertical NAND flash memory devices with alternating Si_3_N_4_/SiO_2_ multilayered stacks require extremely selective etching of Si_3_N_4_ [10,11]. One of the most challenging processes in the integration of these devices is the selective removal of Si_3_N_4_ without thinning the SiO_2_ layer using a high-aspect ratio slit. This is because the memory density of 3D NAND devices increases with the increasing number of Si_3_N_4_/SiO_2_ stack layers [12].

Wet etching is usually employed for etching Si-based semiconductor materials, and this process is still being studied to increase etch selectivity [13,14,15]. However, dry etching is recommended over wet chemical etching owing to its numerous benefits. For instance, the management of dangerous acids and solvents is not required, and anisotropic and isotropic etching profiles can be achieved. Other advantages include excellent resolution, minimal or zero undercutting, consistent outcomes, and improved process control. Plasma etching is a special method for altering the surface characteristics of materials using reactive species produced in the gas phase. While the flux and energy of ions and radicals impact surface-controlled reaction rates, gas-phase chemical reaction rates are determined by plasma parameters such as electron and ion densities and electron temperature [16]. Pinpointing the precise etching surface mechanisms in plasma etching is challenging owing to the complexity of gas-phase reaction pathways, which include excitation, ionization, dissociation, attachment, and recombination. These pathways affect the concentrations of active precursors, including charged and neutral species, and are influenced by the feedstock chemistry, discharge configuration, material properties of the chamber wall, gas and surface temperatures, as well as operating parameters such as pressure and flow rates. In a dry etching reactor, controlling the plasma characteristics is crucial to satisfying the requirements for miniaturized semiconductor device dimensions. Radio frequency (RF) power is used in plasma-based reactive ion etching (RIE) to propel chemical reactions. This process involves a combination of chemical and physical etching [17]. However, RIE procedures are limited by low selectivity, poor chemical/physical balance, and surface damage [18,19]. Efforts have been made to enhance the etch selectivity of Si_3_N_4_ over SiO_2_ in several RIE systems, such as magnetron RIE, RF inductively coupled plasma RIE, and capacitively coupled plasma RIE, using various fluorinated etch gases, including CHF_3_, NF_3_, CH_2_FCH_2_F, and SF_6_. However, the etch selectivity values are small and lie in the range of 0.2–10 and are shown in Table 1. 

To increase the etch selectivity of Si_3_N_4_ over SiO_2_, this study investigated their etching properties, including etch rate and selectivity in NF_3_ and O_2_ gases, using an RIE system. The neutral species produced in the NF_3_ plasma were analyzed using quadrupole mass spectrometry. The surface chemistries of the processed Si_3_N_4_ samples were examined using X-ray photoelectron spectroscopy. Plasma information (e.g., electron temperature and electron density) was calculated using the line-ratio method based on the data obtained using optical emission spectrometry. The effect of adding O_2_ to the NF_3_ plasma on the regulation of etch selectivity was also investigated.

## 2. Materials and Methods

Figure 1 shows the RIE (Rainbow4420, Lam Research, Fremont, CA, USA) equipment used in the experiments. Etching gases, such as NF_3_ and O_2_, were injected into the reaction chamber with an RF discharge. An RF source power of 13.56 MHz in an inductively coupled discharge type ignited a glow plasma to generate the gas-phase etching environment, which comprised positive and negative ions, electrons, radicals, and neutrals from the NF_3_–O_2_ mixture. A coil antenna was used to homogenize the plasma, and an RF bias power of 13.56 MHz at the bottom of the chamber was used to generate a DC self-bias and enhance the energy of the ions bombarding the substrate. The pressure for all experiments was set to 80 mTorr. In most experiments, the bias RF electrode was covered with a 6 in. carrier Si wafer placed on an electrostatic chuck in the reaction chamber. Si_3_N_4_ or SiO_2_ samples of size 1 × 1 in. were glued to the carrier wafer. The samples used in this study were fabricated using plasma-enhanced chemical vapor deposition (PECVD, Precision 5000, Applied Materials, Santa Clara, CA, USA) to a thickness of approximately 300 nm. A helium pressure of 5 Torr was maintained between the surface of the electrostatic chuck and carrier wafer for good heat conduction. For in situ monitoring, a quadrupole mass spectrometer (QMS) (Pfeiffer Vacuum, QMG220F1, Aßlar, Germany), which can detect molecular species of up to 120 amu, was mounted adjacent to the reaction chamber to analyze the plasma-generated species. The ionization region of the mass spectrometer was consistent with that of the reaction chamber. The ionizing electron energy was 30 eV, and the pressure in the mass spectrometer was approximately 70 mTorr. For optical emission experiments of the discharge, a fiberoptic cable was mounted beside the reaction chamber. The spectrograph was an optical multichannel analyzer (Ocean Optics USB 200, Dunedin, FL, USA) covering the spectral range of 200–1100 nm. For the glow discharge of NF_3_/Ar systems, the line ratio method, validated by Zhu et al. [24,25], was used to calculate the electron temperature and electron density from the collected optical emission spectroscopy (OES) data. Argon (2 sccm) was added to the process gas for spectroscopic analysis and obtaining the electron temperature and electron density without affecting the process owing to its inertness. These two parameters were calculated using a reported model [24,25,26].

For ex situ monitoring, spectroscopic ellipsometry (M2000, J. A. Woollam, Lincoln, NE, USA) was used to measure the film thickness before and after etching. Data analysis of the collected spectra was performed using the models for Si_3_N_4_ and SiO_2_. The etch rate was calculated by varying the etching depths of the samples etched for 5 min. The surface chemical properties of the films were analyzed using X-ray photoelectron spectroscopy (XPS; Multilab 2000, VG Scientific, London, UK) with a twin anode X-ray source for Mg–Kα.

## 3. Results

In RIE, material etching involves chemical and physical etching [17]; the former is involved in the reaction of neutrals (e.g., free radicals) with the surface of the etched material to produce volatile products, whereas the latter is involved in the energetic ion bombardment for ejecting surface atoms [17,27,28]. The result of the etching process is influenced by numerous parameters, such as the etching gas, generator frequency, flow rate, pressure, power, and electrode geometry [29]. However, owing to the limitations of our equipment, only the source and bias powers and the components of the etching gas mixture were investigated. The source and bias powers were studied to control the energy of the ions bombarding the wafer during anisotropic profile creation and etch rate enhancement [28,30,31,32,33,34,35]. The etching gas ratios of NF_3_ and O_2_ were studied to clarify the surface reactions that occurred during etching. The chamber pressure and total gas flow were maintained at 80 mTorr and 100 sccm, respectively.

### 3.1. Effect of Source and Bias Powers on Etch Rate and Etch Selectivity

Two types of experiments were performed to determine the influence of power on the etching characteristics of both materials. In the first experiment, the effect of source power on Si_3_N_4_ and SiO_2_ etching was investigated (Figure 2). The etch rates of both materials increase as the source power increases; however, Si_3_N_4_ exhibits higher etch rates at all source powers. The etch selectivity increases by a factor greater than three. Particularly, the maximum etch selectivity exhibited was 24 at a source power of 30 W. The etch selectivity values of Si_3_N_4_ and SiO_2_ decrease with increasing source power (Figure 2b).

In the second experiment, the dependence of the etching characteristics of Si_3_N_4_ and SiO_2_ on the bias power was studied. The power source was maintained at 300 W, and the experimental parameters were similar to those in the previous experiment. Figure 2c shows that the etch rates of Si_3_N_4_ and SiO_2_ are increasing with increasing bias power. The Si_3_N_4_ etch rate was higher than the SiO_2_ etch rate by a factor of 1.5. As shown in Figure 2d, an increase in the bias power resulted in a gradual decrease in the etch selectivity from 3.4 to 1.8.

Figure 2a,c indicate that the bias power has a stronger effect on the measured etch rate than the source power, considering the same total power. However, the etch selectivity decreases with increasing source or bias power. Based on the investigated power conditions, the maximum selectivity of approximately 24 was obtained at applied bias and source powers of 0 and 30 W, respectively. 

XPS described the surface modifications of the materials after etching. In Figure 3, the experimental spectra are shown as solid lines, and the fitted spectra are depicted as red and pale violet peaks. The modified surfaces of Si_3_N_4_ samples at various source powers are shown in Figure 3a. The Si (2p) peak at an applied source power of 30 W shifts from 101.2 to 103.3 eV, which can be reportedly attributed to the SiO_x_N_y_ peak [36,37]. In contrast to the etched Si_3_N_4_ surface at a source power of 30 W, no peak shift was observed on the etched surface when the source power increased from 100 to 500 W. In Figure 3b, no shift is observed in the Si (2p) XPS spectrum of the Si_3_N_4_ surface after etching. Hence, the surface reaction effect on Si_3_N_4_ etching was not significant. The SiO_x_N_y_ peak shown in Figure 3a indicates that the Si_3_N_4_ surface was oxidized after etching at applied bias and source powers of 0 and 30 W, respectively. 

Mass spectrometry was used to determine the relative changes in the concentrations of reactive neutral species generated in the plasma at high power. Figure 4a shows the correlations between the mass spectrometric signal intensities of O, F, NO, and NF_2_ and the source power. The concentration of the species slightly depended on the increase in source power. Previous studies have used NO as an important precursor for etching Si_3_N_4_ owing to its ability to remove N atoms from the surfaces of etched materials [38,39,40,41]. The detection of NO in the plasma indicates that Si_3_N_4_ etching occurred via surface reactions. Figure 4b shows that the concentration of species remained nearly constant with an increasing bias power, which is similar to the data in Figure 4a. This result can be attributed to the low dissociation energy of NF_3_ owing to the electron impact in the plasma [42,43]. As power increases, the negligible increase in the concentration of the species related to surface reactions during etching indicates that chemical etching is not the dominant factor for the increase in etch rates, as shown in Figure 2a,c.

Electron temperature (T_e_) and electron density (n_e_) are the most important parameters in plasma etching. They are used to determine plasma characteristics, such as the plasma potential (ion bombardment energy on a substrate), sheath thickness, electron thermal velocity, ion flux, and etch rate related to physical etching. In this study, T_e_ and n_e_ were calculated using the equations validated by Zhu et al. [21,22,23]. Figure 4c,d show the relation between the applied power and T_e_ and n_e_, as denoted by black squares and blue rectangles, respectively. Figure 4c shows that T_e_ increases with increasing source power, and the electron density for the source power of 30–300 W is in the range 0.16 × 10^11^–12.65 × 10^11^ cm^−3^. T_e_ and n_e_ represent the plasma characteristics, which lead to a particular behavior related to the generation of plasma species and the energy of ion bombardment on the etched surface. Therefore, the increase in T_e_ and n_e_ increases the ion density and ion bombardment energy, thereby enhancing the etch rates of both materials, as shown in Figure 2a. Figure 4d shows the relation between the bias power and T_e_ and n_e_, which remain nearly constant. In contrast to the source power, the bias power only increases the ion energy (i.e., DC bias) [44,45]. Consequently, the increase in etch rate shown in Figure 2c is positively correlated with the ion energy.

Considering the relation between the etch rate and applied power (Figure 2a,c), the bias power has a stronger positive effect on the etch rate than the source power.

### 3.2. Effect of the O_2_/NF_3_ Gas Ratio on Etch Rates and Etch Selectivity

Figure 5 shows the effect of O_2_ addition on the etching of Si_3_N_4_ and SiO_2_. Figure 5a shows that the Si_3_N_4_ etch rate significantly enhances with an increasing O_2_/NF_3_ ratio. In all cases, the etch rates of Si_3_N_4_ exceed those of SiO_2_ by a factor of 10 or more. However, an increase in the SiO_2_ etch rate decreases the etch selectivity from 24 to 10, corresponding to an increase in the O_2_/NF_3_ ratio (shown in Figure 5b).

The relationship between the intensity of the oxidized SiO_x_N_y_ peak of the etched Si_3_N_4_ and the variation in gas ratio is described by the Si (2p) spectra of the Si_3_N_4_ surface after etching, as shown in Figure 6a. The Si (2p) peak increased the SiO_x_N_y_ peak intensity at high binding energies when the gas ratio increased from 0.1 to 1. This corresponded to an increased number of Si atoms bonded to O atoms during etching. However, the SiO_2_ peak did not shift when the gas ratio increased (Figure 6b). Figure 6c shows a significant increase in the O (1s) intensity and a simultaneous decrease in the N (1s) intensity with increasing gas ratio.

As shown in Figure 7a, the NO intensity measured using the QMS increased linearly with the gas ratio. The results shown in Figure 6a–c and Figure 7a indicate that the etching of Si_3_N_4_ at various O_2_/NF_3_ gas ratios and applied bias and source powers of 0 and 30 W, respectively, was carried out predominantly by the surface reactions of the NO species. 

Figure 7b shows the dependencies of T_e_ and n_e_ on the gas ratio; the former is nearly constant at 2.5 ± 0.1 eV for all investigated ratios, whereas the latter increases slightly with increasing gas ratio. Therefore, the increase in etch rate at various gas ratios does not correlate with the energy of the ions bombarding the etched surface.

## 4. Discussion

The effect of power increment is crucial in enhancing the etch rates of both materials (Figure 2); however, it has an insignificant effect on etch selectivity. High etch selectivity was not observed at high applied powers because ion-driven etching was dominant (Figure 4). Surface oxidation was only observed at applied bias and source powers of 0 and 30 W, respectively; it was not observed at higher powers (Figure 3a,b). In Figure 3a, the shifted SiO_x_N_y_ peak of Si_3_N_4_ etching indicates that the highest selective etching at the aforementioned powers was achieved using plasma neutrals. The high selectivity of Si_3_N_4_ over SiO_2_ at low power can be attributed to the chemical reactions occurring only on the surface of Si_3_N_4_. Previous studies have reported that for etching at low energies, adsorption preceded a chemical reaction rather than physical etching [46,47,48]. Our results were consistent with these conclusions.

The chemistry of the etching process was investigated by varying the gas ratios. Figure 6a shows the surface oxidation in the etched Si_3_N_4_ with a significant increase in the O (1s) intensity (Figure 6c), which was not observed in the SiO_2_ etching process (Figure 6b). Oxidation is related to the NO generated by the plasma containing NF_3_–O_2_; therefore, the enhanced Si_3_N_4_ etch rate in Figure 5 is caused by an increase in NO intensity, as shown in Figure 7a. The nearly constant plasma information (T_e_ and n_e_) with changes in the gas ratio in Figure 7b indicates that physical etching is not dominant. The increase in the etch rate with increasing gas ratios did not significantly correlate with physical etching. 

Based on the results of this study and previous reports, the mechanism of Si_3_N_4_ etch in the plasma containing NF_3_–O_2_ can be explained as follows: nitrogen trifluoride is easily dissociated by electron impact. Because the NF_2_–F bond strength (2.5 eV) is lower than the electron affinity of the F atom (3.6 eV), the threshold energy for the first step, that is, dissociative electron attachment, is approximately zero [42,43].
e^−^ + NF_3_ → NF_2_ +F^−^(1)

In the presence of O_2_, oxygen atoms quickly react with NF_2_ and NF as follows [49,50]:O + NF_2_ → NF + OF(2)
O + NF → NO + F(3)

NO is produced through plasma containing nitrogen and oxygen, as follows [51]:N + O + (M) → NO* + (M)(4)
N^+^ + O_2_ → O^+^ + NO + 2.3 eV (about 10%)(5)
N^+^ + O_2_ → O + NO^+^ + 6.7 eV (about 86%)(6)
N^+^ + O_2_ → O + NO^+^* + 0.3 eV (about 4%)(7)

Several mechanisms of Si_3_N_4_ etching have been reported [38,39,51]. They are all related to the bonding of a surface N atom with NO, which is highly reactive owing to its unpaired electron that is localized around the N atom. Volatile species are produced in the subsequent reaction steps. Therefore, the NO molecules enhance the rate at which N atoms are removed from the surface. The XPS results in Figure 6 reveal a close correlation between the removal of surface N atoms and the increased NO density with the increasing O_2_/NF_3_ gas ratio (Figure 7a). The nitrogen content of the Si_3_N_4_ surface is inversely proportional to the NO density (Figure 6c and Figure 7a). The proposed mechanisms are as follows:N_surface_ + NO_gas_ → N_2 gas_ + O_surface_(8)
N_surface_ + NO_gas_ → N_2_O_gas_(9)
2 N_surface_ + 2 NO_gas_ → 2 N_2 gas_ + O_2 gas_(10)

Our results are consistent with the mechanism proposed in Equation (8). It is first-order in NO, and each N atom removed from the surface is replaced with an O atom, as shown in Figure 6c. This agrees with the linear dependencies of the etch rates and surface oxidation on the NO density (Figure 2c and Figure 7a). The mechanism in Equation (9) cannot explain the increased surface oxidation, as shown in Figure 6c, because N_2_O was not detected by the QMS in our experiments. The third mechanism, shown in Equation (10), does not involve Si_3_N_4_ surface oxidation, as shown in the XPS spectra in Figure 3 and Figure 6. The depletion of N on the surface by NO (Equation (8)) led to a reaction between the Si and F atoms, yielding the by-product SiF_4_ as follows:4F + Si_surface_ → SiF_4_(11)

The oxidation of NO occurring on the Si_3_N_4_ surface and the etch process are illustrated in Figure 8. The decrease in the selectivity of Si_3_N_4_ over SiO_2_ with an increasing ratio of O_2_/NF_3_ is caused by an increase in the SiO_2_ etch rate (shown in Figure 4c), which can be explained by the increased F concentration in the plasma with decreasing O_2_ content [52].

## 5. Conclusions

This study investigated the characteristics of etching Si_3_N_4_ and SiO_2_ at various source and bias powers and O_2_/NF_3_ gas ratios using RIE equipment. The etch rates of both materials increased with increasing power. Compared with previous studies, a high etch selectivity of 24 was achieved at low power [6,20,21,22,23]. The high selectivity of Si_3_N_4_ over SiO_2_ was caused by the creation of SiO_x_N_y_ bonds on the Si_3_N_4_ surface from NO oxidation. In the simultaneous etching of Si_3_N_4_ and SiO_2_ to increase the Si_3_N_4_ etch selectivity, NO-related chemical etching should be considered. Contrastingly, RF power must be controlled to enhance the etch rates of the materials. However, this study has not addressed the decreasing etch selectivity with the increasing SiO_2_ etch rate and O_2_/NF_3_ gas ratio. Therefore, the relationship between O_2_ concentration and increasing SiO_2_ etch rate requires further investigation.

## Figures and Tables

**Figure 1 sensors-24-03089-f001:**
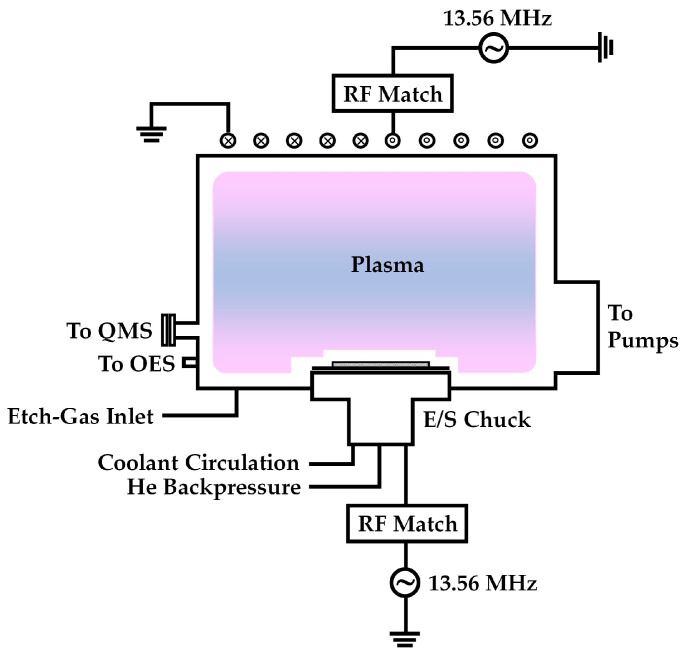
Schematic of the implemented RIE.

**Figure 2 sensors-24-03089-f002:**
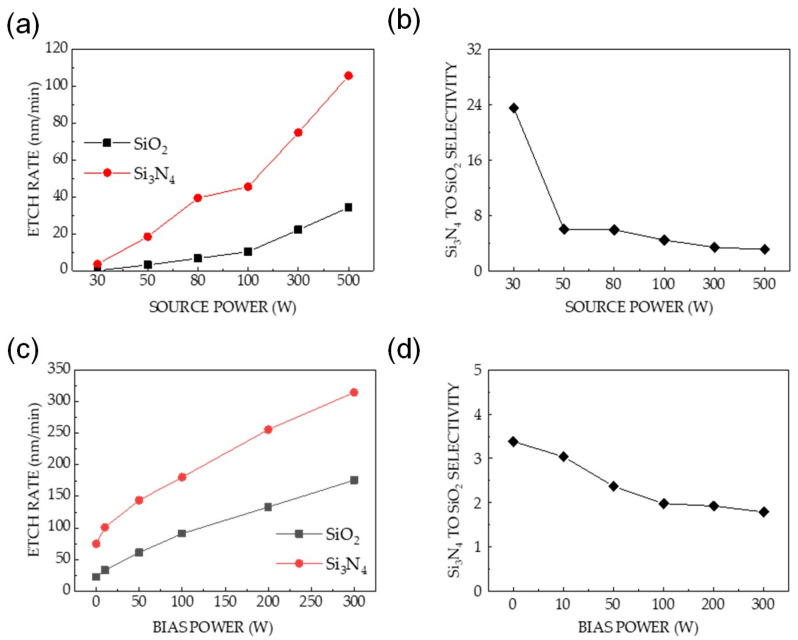
Etching characteristics of Si_3_N_4_ and SiO_2_ at a reactor pressure of 80 mTorr and a gas mixture of O_2_, NF_3_, and Ar with gas flow rates of 9, 89, and 2 sccm, respectively. At an applied bias power of 0 W: (**a**) etch rates of Si_3_N_4_ and SiO_2_ as a function of source power and (**b**) Si_3_N_4_-to-SiO_2_ selectivity calculated using the etch rate ratio. At an applied source power of 300 W: (**c**) etch rates of Si_3_N_4_ and SiO_2_ as a function of bias power and (**d**) Si_3_N_4_-to-SiO_2_ selectivity.

**Figure 3 sensors-24-03089-f003:**
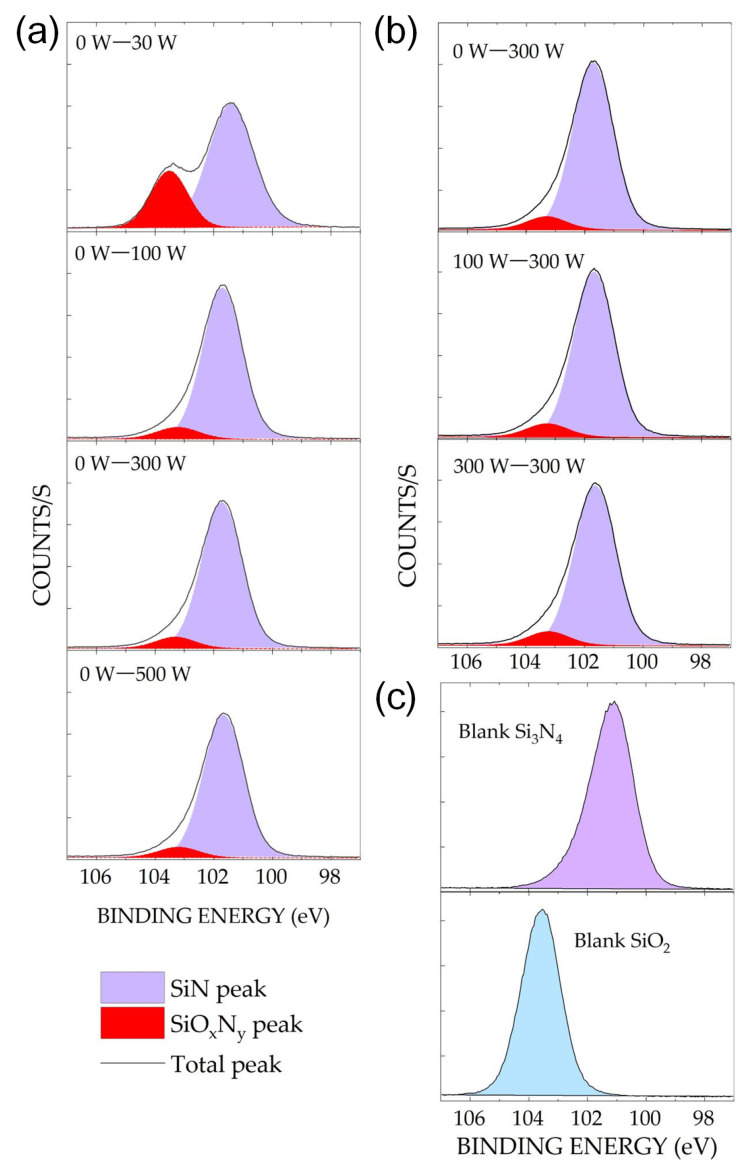
Si (2p) XPS spectra of the Si_3_N_4_ surface after etching at various powers: (**a**) applied bias power = 0 W and source power = 30–500 W; (**b**) applied source power = 300 W and bias power = 0–300 W. The measurements were performed at a pressure of 80 mTorr and a gas mixture of O_2_, NF_3_, and Ar with gas flow rates of 9, 89, and 2 sccm, respectively; (**c**) XPS spectra of the blank Si_3_N_4_ or SiO_2_ wafer sample.

**Figure 4 sensors-24-03089-f004:**
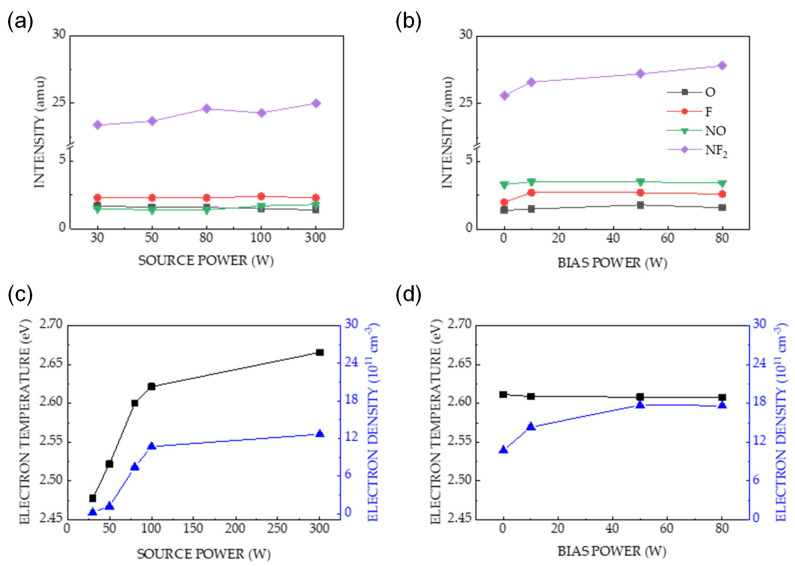
Dependence of the mass spectrometric signal intensities of O, F, NO, and NF_2_ on power: (**a**) applied bias power = 0 W and source power = 30–300 W; (**b**) applied source power = 100 W and bias power = 0–80 W. The measurements were performed at a pressure of 80 mTorr and a gas mixture of O_2_, NF_3_, and Ar with gas flow rates of 9, 89, and 2 sccm, respectively. The electron temperature and electron density of the mixed gas plasma of NF_3_ and O_2_ are represented as a function of the power: (**c**) applied bias power = 0 W and source power = 30–300 W; (**d**) applied source power = 100 W and bias power = 0–80 W. The measurements were performed at a pressure of 80 mTorr and a gas mixture of O_2_, NF_3_, and Ar with gas flow rates of 9, 89, and 2 sccm, respectively.

**Figure 5 sensors-24-03089-f005:**
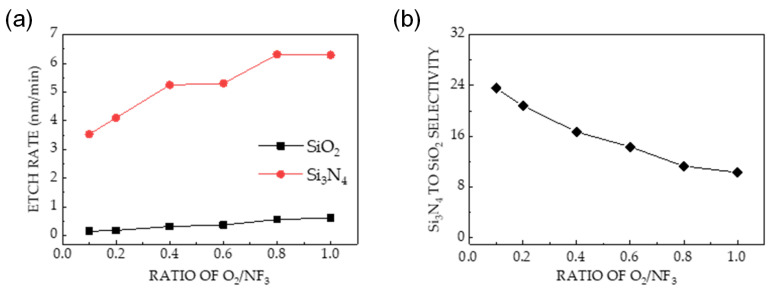
Etch characteristics of Si_3_N_4_ and SiO_2_ at a pressure, applied bias power, and applied source power of 80 mTorr, 0 W, and 30 W, respectively. (**a**) Etch rates of Si_3_N_4_ and SiO_2_ as a function of the O_2_/NF_3_ ratio. (**b**) Si_3_N_4_-to-SiO_2_ selectivity calculated using the etch rate ratio.

**Figure 6 sensors-24-03089-f006:**
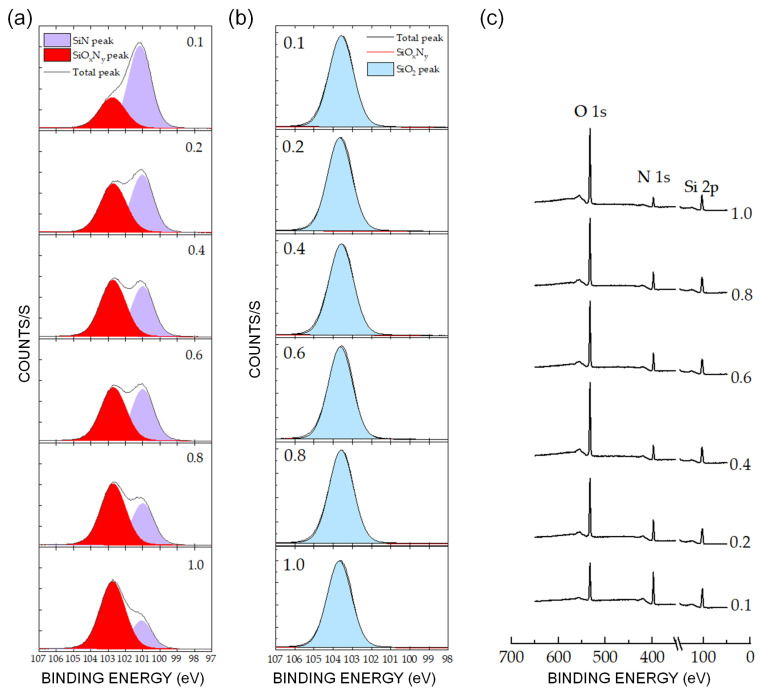
Si (2p) XPS spectra of (**a**) Si_3_N_4_ and (**b**) SiO_2_ surfaces after etching at an applied bias power of 0 W, an applied source power of 30 W, a pressure of 80 mTorr, and various gas ratios of O_2_/NF_3_. The chemical shift is observed from 101 eV in Si_3_N_4_ to 102.7 eV in SiO_x_N_y_. (**c**) O (1s) and N (1s) photoelectron line intensities as a function of O_2_/NF_3_ after etching the silicon nitride surface at the aforementioned parameters.

**Figure 7 sensors-24-03089-f007:**
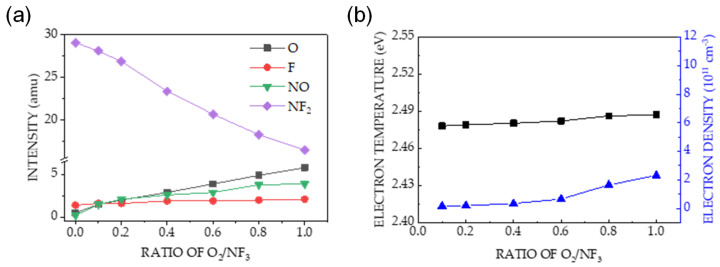
(**a**) Dependence of mass spectrometric signal intensities of O, F, NO, and NF_2_ on the ratio of O_2_/NF_3_. The measurements were performed at a pressure, applied bias power, and applied source power of 80 mTorr, 0 W, and 30 W, respectively. (**b**) Dependence of electron temperature and electron density on the gas ratio of O_2_/NF_3_ at the aforementioned parameters.

**Figure 8 sensors-24-03089-f008:**
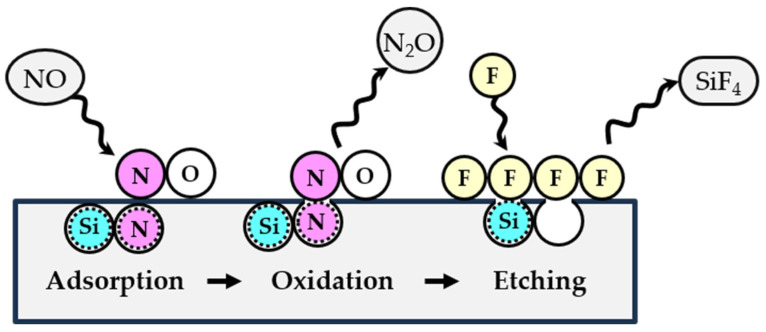
The scheme of the possible reaction occurring on the Si_3_N_4_ surface in the etch process.

**Table 1 sensors-24-03089-t001:** Selectivity of Si_3_N_4_ to SiO_2_ in some published studies.

Gas Mixture	Reactor	Si_3_N_4_ Selectivity	Ref.
CHF_3_	Radio-frequency inductively coupled plasma	1	[6]
C_2_F_6_		0.8	
C_3_F_6_		0.5	
C_3_F_6_, H_2_		0.2	
CHF_3_	Radio-frequency inductively coupled plasma	1.3	[20]
C_4_F_8_		0.4	
NF_3_, C_2_H_4_	Magnetically confined inductively coupled plasma	0.75	[21]
C_5_F_8_, O_2_	Capacitively coupled plasma	10	[22]
SF_6_, H_2_	Magnetic neutral loop discharge plasma	2.4	[23]
CH_2_FCH_2_F, O_2_, Ar		3.8	

## Data Availability

Data are contained within the article.

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
