# Peer review of "Enhancing Si3N4 Selectivity over SiO2 in Low-RF Power NF3–O2 Reactive Ion Etching: The Effect of NO Surface Reaction"

_sensors, 2024, doi:10.3390/s24103089_

Round 1
Reviewer 1 Report
Comments and Suggestions for Authors
The manuscript focuses on investigating "Enhancing Si3N4 Selectivity over SiO2 in Low-RF Power NF3-O2 Reactive Ion Etching: The Effect of NO Surface Reaction." The parameters affecting the etch rate were systematically studied, making it potentially interesting for readers. I would recommend publishing it after minor revisions.
Comments:
1. The architecture of Si3N4-SiO2 could be presented more clearly through well-labeled diagrams for the readers' better understanding.
2. The discrepancy in the binding energy of SiON between Figure 3a (>103 eV) and Figure 6a (102.7 eV) needs clarification.
3. What is the ionic form of the plasma O2/NF3, and does it involve active oxygen? How does the formation of NO occur with active oxygen, leading to the etching of nitrogen?
4. The concept of selective etching might be pseudo because nitrogen and oxygen could be etched simultaneously and then replaced by active oxygen.
5. Equations 4-6 should be further validated through additional characterizations. The etching of nitrogen should involve active nitrogen easily combining with active oxygen from the plasma gas.
6. The mechanism of selective etching should be clearly illustrated in a figure for the readers' benefit.
Author Response
Q1. The architecture of Si3N4-SiO2 could be presented more clearly through well-labeled diagrams for the readers' better understanding.
⇒A1. Thank you for your query. We used the Si3N4 / SiO2 wafer in this study, not the Si3N4-SiO2 wafer. Therefore, the well-labeled diagram of the Si3N4-SiO2 architecture cannot be added to the manuscript. We changed “Si3N4 and SiO2 samples of size 1 × 1 in. were …” to “Si3N4 or SiO2 samples of size 1 × 1 in. were …” in the revised manuscript to avoid the readers’ misunderstanding.
⇒(Page 3, Paragraph 1). Figure 1 shows the RIE (Rainbow4420, Lam Research, United States) equipment used in the experiments. Etching gases, such as NF3 and O2, were injected into the reaction chamber with an RF discharge. An RF source power of 13.56 MHz in an induc-tively-coupled-discharge type ignited a glowing plasma to generate the gas-phase etching environment, which comprised positive and negative ions, electrons, radicals, and neutrals, from the NF3–O2 mixture. A coil antenna was used to homogenize the plasma, and an RF bias power of 13.56 MHz at the bottom of the chamber was used to generate a DC self-bias and enhance the energy of the ions bombarding the substrate. The pressure for all experiments was set to 80 mTorr. In most experiments, the bias RF electrode was covered with a 6-in. carrier Si wafer placed on an electrostatic chuck in the reaction chamber. Si3N4 or SiO2 samples of size 1 × 1 in. were glued to the carrier wafer.
Q2. The discrepancy in the binding energy of SiON between Figure 3a (>103 eV) and Figure 6a (102.7 eV) needs clarification.
⇒A2. Thank you for your insightful question. We used separate samples to study the etching process of Si3N4. Therefore, the discrepancy of the binding energy (~ 0.3 eV) between Figure 3(a) and Figure 6(a) relates to the surface properties of each Si3N4 sample (the chemical nature of the neighboring atoms, oxidation state, and local chemical environment of Si 2p, etc.).
Q3. What is the ionic form of the plasma O2/NF3, and does it involve active oxygen?
⇒A3-1. Thank you for raising this crucial question. The species in the O2/NF3 plasma relating to the etch process are NF2, O, NF, F, and NO. We described these species on page 9, paragraph 4.
⇒(Page 9, Paragraph 4). Based on the results of this study and previous reports, the mechanism of Si3N4 etch in the plasma containing NF3–O2 can be explained as follows: Nitrogen trifluoride is easily dissociated by electron impact. Because the NF2–F bond strength (2.5 eV) is lower than the electron affinity of the F atom (3.6 eV), the threshold energy for the first step, i.e., dissociative electron attachment, is approximately zero [42,43].
e- + NF3 " NF2 +F- (1)
In the presence of O2, oxygen atoms quickly react with NF2 and NF as follows [49,50]:
O + NF2 " NF + OF (2)
O + NF " NO + F (3)
How does the formation of NO occur with active oxygen, leading to the etching of nitrogen?
⇒A3-2. Thank you for your insightful question. Formation of NO occurs as follows;
|
e- + NF3 ¦ NF2 +F- |
(1) |
|
O + NF2 ¦ NF + OF |
(2) |
|
O + NF¦ NO + F |
(3) |
|
|
|
Q4. The concept of selective etching might be pseudo because nitrogen and oxygen could be etched simultaneously and then replaced by active oxygen.
⇒A4. Thank you for your important question. As we know, bond strengths for Si-F and Si-O are 132 kcal/mol and 111 kcal/mol, respectively. Therefore, after nitrogen or oxygen is etched, fluorine radicals in the plasma will react with Si to create SiF4 while the reaction of active oxygen with Si is difficult to occur.
Q5. Equations 4-6 should be further validated through additional characterizations. The etching of nitrogen should involve active nitrogen easily combining with active oxygen from the plasma gas.
⇒A5. We thank the reviewer’s critical and helpful comments. We agree with the reviewer and added “NO is produced through plasma …” on page 9, paragraph 4.
⇒(Page 9, Paragraph 4). Based on the results of this study and previous reports, the mechanism of Si3N4 etch in the plasma containing NF3–O2 can be explained as follows: Nitrogen trifluoride is easily dissociated by electron impact. Because the NF2–F bond strength (2.5 eV) is lower than the electron affinity of the F atom (3.6 eV), the threshold energy for the first step, i.e., dissociative electron attachment, is approximately zero [42,43].
|
e- + NF3 ¦ NF2 +F- |
(1) |
In the presence of O2, oxygen atoms quickly react with NF2 and NF as follows [49,50]:
|
O + NF2 ¦ NF + OF |
(2) |
|
O + NF¦ NO + F |
(3) |
NO is produced through plasma containing nitrogen and oxygen [51]
|
N + O + (M) ¦ NO* + (M) |
(4) |
|
N+ + O2 ¦ O+ + NO + 2.3 eV (about 10%) |
(5) |
|
N+ + O2 ¦ O + NO+ + 6.7 eV (about 86%) |
(6) |
|
N+ + O2 ¦ O + NO+* + 0.3 eV (about 4%) |
(7) |
Q6. The mechanism of selective etching should be clearly illustrated in a figure for the reader's benefit.
⇒A6. We thank the reviewer’s helpful comments. We agree with the comment and added Figure 8 to page 10, paragraph 2.
Figure 8. The scheme of the possible reaction occurring on the Si3N4 surface in the etch process.

Reviewer 2 Report
Comments and Suggestions for Authors
This work systematically studied the influence of source power, bias power, and O2/NF3 ratio on the etching rate and selectivity of Si3N4 and SiO2. The author claimed that the high selectivity etching of Si3N4 over SiO2 originates from the surface oxidation by NO. This work provides a fundamental understanding of etching processes, and I would suggest acceptance of this work after a major revision. Hope my comments can further enhance the manuscript.
1. Figure 2(a), the author claimed that “The etch rates of both materials increase linearly with the source power, however…”. However, it does not look like a linear increase. The etching rate becomes much faster at high source power.
2. XPS spectra of pristine Si3N4 and SiO2 should be provided.
3. In figure 3(a), the SiN peak at 0W-30W appears at ~101.4 eV, and the peak shifts to ~101.8 eV at 0-100W, and then keeps constant at that position. Does the author have explanations for that? SiON peak in Figure 3(a) is located at ~ 103.5 eV, this peak position should be assigned to SiO2.
4. Peak positions for 0W-30W in Figure 3(a) should be the same as that for O2/NF3 = 0.1 in Figure 6(a), however, there are about 0.4 eV energy shifts. Why is that?
5. Page 5, line 163. “the experimental spectra are shown as solid lines and simulated spectra are depicted as red and pale violet peaks.” The words, “simulated spectra”, are not proper. Using “fitted spectra” would be better.
Author Response
Q1. Figure 2(a), the author claimed that “The etch rates of both materials increase linearly with the source power, however…”. However, it does not look like a linear increase. The etching rate becomes much faster at high source power.
⇒A1. We thank the reviewer’s critical and helpful comments. We agree with the reviewer and modified on page 4, paragraph 3.
⇒Page 4, Paragraph 3). Two types of experiments were performed to determine the influence of power on the etching characteristics of both materials. In the first experiment, the effect of source power on Si3N4 and SiO2 etching was investigated (Figure 2). The etch rates of both materials increase as the source power increases; however, Si3N4 exhibits higher etch rates at all source powers. The etch selectivity increases by a factor greater than three. Particularly, the maximum etch selectivity exhibited was 24 at a source power of 30 W. The etch selectivity values of Si3N4 and SiO2 decrease with increasing source power (Figure 2b).
Q2. XPS spectra of pristine Si3N4 and SiO2 should be provided.
èA2. We thank the reviewer’s valuable comments. We agree with the reviewer and added the XPS spectra of the blank Si3N4 and SiO2 wafer samples into Figure 3c, page 5.
Figure 3. Si (2p) XPS spectra of the Si3N4 surface after etching at various powers: (a) applied bias power = 0 Wand source power = 30–500 W; (b) applied source power = 300 W and bias power = 0–300 W. The measurements were performed at a pressure of 80 mTorr and a gas mixture of O2, NF3, and Ar with gas flow rates of 9, 89, and 2 sccm, respectively; (c) XPS spectra of the blank Si3N4 or SiO2 wafer sample.
Q3. In Figure 3(a), the SiN peak at 0 W-30 W appears at ~101.4 eV, and the peak shifts to ~101.8 eV at 0-100 W, and then keeps constant at that position. Does the author have explanations for that?
⇒A3-1. We thank the reviewer’s helpful comments. We used separate samples to study the etching process of Si3N4. Therefore, the binding energy shift in Figure 3a is related to the surface properties of each Si3N4 sample (the chemical nature of the neighboring atoms, oxidation state, and local chemical environment of Si 2p, etc.).
SiON peak in Figure 3(a) is located at ~ 103.5 eV, this peak position should be assigned to SiO2.
⇒A3-2. Thank you for raising this crucial question. SiON includes different compositions depending on the ratio of N and O atoms on the Si3N4 surface (e.g.: SiO3N, SiO2N2, SiON3, etc.). If SiON has a larger number of O atoms than N atoms, its binding energy will increase. In Figure 3(a), the binding energy of the SiON peak, located at 103.3 eV, is nearly equal to the binding energy of SiO2 (103.6 eV). It indicates that the number of O atoms in SiON is larger than that of N atoms. We changed SiON to SiOxNy in the whole manuscript to explain SiON more clearly.
Q4. Peak positions for 0 W-30 W in Figure 3(a) should be the same as that for O2/NF3 = 0.1 in Figure 6(a), however, there are about 0.4 eV energy shifts. Why is that?
⇒A4. We thank the reviewer’s valuable comments. Similar to the answer of Q3, the discrepancy of the binding energy (~ 0.4 eV) between Figure 3(a) and Figure 6(a) relates to the surface properties of each Si3N4 sample (the chemical nature of the neighboring atoms, oxidation state, and local chemical environment of Si 2p, etc.).
Q5. Page 5, line 163. “The experimental spectra are shown as solid lines and simulated spectra are depicted as red and pale violet peaks.” The words, “simulated spectra”, are not proper. Using “fitted spectra” would be better.
⇒A5. We thank the reviewer’s critical and helpful comments. We agree with the comment and modified “simulated spectra” with “fitted spectra” in the revised manuscript.
⇒(Page 5, Paragraph 3). XPS described the surface modifications of the materials after etching. In Figure 3, the experimental spectra are shown as solid lines, and fitted spectra are depicted as red and pale violet peaks. The modified surfaces of Si3N4 samples at various source powers are shown in Figure 3a. The Si (2p) peak at an applied source power of 30 W shifts from 101 to 102.7 eV, which can be reportedly attributed to the SiOxNy peak [36,37]. In contrast to the etched Si3N4 surface at a source power of 30 W, no peak shift was observed on the etched surface when the source power increased from 100 to 500 W. In Figure 3b, no shift is observed in the Si (2p) XPS spectrum of the Si3N4 surface after etching. Hence, the surface reaction effect on Si3N4 etching was not significant.

Reviewer 3 Report
Comments and Suggestions for Authors
Title: Enhancing Si3N4 Selectivity over SiO2 in Low-RF Power NF3–O2 Reactive Ion Etching: The Effect of NO Surface Reaction
The authors of the paper entitled: “Enhancing Si3N4 Selectivity over SiO2 in Low-RF Power NF3–O2 Reactive Ion Etching: The Effect of NO Surface Reaction” investigated the etching selectivity of Si3N4 and SiO2 in an NF3/O2 radio-frequency glow discharge. The etch rate linearly depended on the source and bias powers, whereas the etch selectivity was affected by the power and ratio of the gas mixture. They found that the selectivity can be controlled by lowering the power with a suitable gas ratio which affect the surface reaction during the etching process.
The paper is well written and can be considered for publication in Sensors after addressing the following issues:
Major Revisions:
1) The solid surfaces of Si3N4 and SiO2 must be characterized by additional physico-chemical techniques in order to understand the impact on the morphology and particles. Additionally, the authors can concluded the effect of the oxidation and nitrogen reduction. The XPS alone cannot confirm the conclusion of this paper
2) Most recent published in this field should be cited.
3) A comparative table should be placed in the introduction section to cite all other oxides/materials that used for this application. It helps readers to understand the novel insights of the authors.
These major revisions must be thoroughly addressed before the paper can be considered for publication in the Journal Sensors.
Author Response
Q1. The solid surfaces of Si3N4 and SiO2 must be characterized by additional physico-chemical techniques to understand the impact on the morphology and particles. Additionally, the authors can conclude the effect of oxidation and nitrogen reduction. The XPS alone cannot confirm the conclusion of this paper
⇒A1. We thank the reviewer’s critical and helpful comments. We agree with the reviewer. But in this work, XPS can give useful information like peak shift, change atom intensity, and surface oxidation and this information is enough for the conclusion of this manuscript.
Q2. The most recent published in this field should be cited.
⇒A2. We thank the reviewer’s valuable comments. We agree with the comment and cited recent published studies in the revised manuscript.
⇒(Page 2, Paragraph 2). Wet etching is usually employed for etching Si-based semiconductor materials and this process is still being studied to increase etch selectivity [13-15]. However, dry etching is recommended over wet chemical etching owing to its numerous benefits. For instance, the management of dangerous acids and solvents is not required and aniso-tropic and isotropic etching profiles can be achieved.
References
- Song, J.; Park, K.; Jeon, S.; Lee, J.; Kim, T. Development of a novel wet cleaning solution for Post-CMP SiO2 and Si3N4 Mater. Sci. Semicond. Process. 2022, 140, 10635
- Zhou, Z.; Han, S.; Wu, Y.; Hang, T.; Ling, H.; Guo, J.; Wang, S.; Li, M. Study on the SiO2 wet-etching mechanism using γ-ureidopropyltriethoxysilane as an inhibitor for 3D NAND fabrication. ACS Appl. Electron. Mater. 2024, 6, 2788–2795.
- Kim, T.; Park, T.; Lim, S. Improvement of Si3N4/SiO2 etching selectivity through the passivation of SiO2 surface in aromatic carboxylic acid-added H3PO4 solutions for the 3D NAND integration. Surf. Sci. 2023, 619, 156758.
Q3. A comparative table should be placed in the introduction section to cite all other oxides/materials that used for this application. It helps readers to understand the novel insights of the authors.
⇒A3. We thank the reviewer’s critical and helpful comments. We agree with the reviewer and added the comparative table on page 2.
Table 1. Selectivity of Si3N4 to SiO2 in some published studies.
|
Gas mixture |
Reactor |
Si3N4 Selectivity |
Ref. |
|
CHF3 |
Radio-Frequency Inductively Coupled Plasma |
1 |
[6] |
|
C2F6 |
0.8 |
||
|
C3F6 |
0.5 |
||
|
C3F6, H2 |
0.2 |
||
|
CHF3 |
Radio-Frequency Inductively Coupled Plasma |
1.3 |
[20] |
|
C4F8 |
0.4 |
||
|
NF3, C2H4 |
Magnetically Confined Inductively Couple Plasma |
0.75 |
[21] |
|
C5F8, O2 |
Capacitively Coupled Plasma |
10 |
[22] |
|
SF6, H2 |
Magnetic Neutral Loop Discharge Plasma |
2.4 |
[23] |
|
CH2FCH2F, O2, Ar |
3.8 |

Round 2
Reviewer 2 Report
Comments and Suggestions for Authors
My concerns have been addressed. I would suggest acceptance of the revised version.
Reviewer 3 Report
Comments and Suggestions for Authors
I recommend the publication of the corrected version.